# MODULE-AWARE PARAMETER-EFFICIENT MACHINE UNLEARNING ON TRANSFORMERS

## ABSTRACT

Transformer has become fundamental to a vast series of pre-trained large models that have achieved remarkable success across diverse applications. Machine unlearning, which focuses on efficiently removing specific data influences to comply with privacy regulations, shows promise in restricting updates to influence-critical parameters. However, existing parameter-efficient unlearning methods are largely devised in a module-oblivious manner, which tends to inaccurately identify these parameters and leads to inferior unlearning performance for Transformers. In this paper, we propose `MAPE-Unlearn`, a module-aware parameter-efficient machine unlearning approach that uses a learnable pair of masks to pinpoint influence-critical parameters in the heads and filters of Transformers. The learning objective of these masks is derived by desiderata of unlearning and optimized through an efficient algorithm featured by a greedy search with a warm start. Extensive experiments on various Transformer models and datasets demonstrate the effectiveness and robustness of `MAPE-Unlearn` for unlearning.

## 1 INTRODUCTION

Transformer architecture (Vaswani et al. (2017)) has achieved superior performance in the field of natural language processing. Its models, e.g., BERT (Devlin et al. (2018)) and GPT (Achiam et al. (2023)), show impressive performance in a wide range of downstream tasks (Wei et al. (2021); Hao et al. (2019)). In light of privacy regulations, such as General Data Protection Regulation (GDPR) (Hoofnagle et al. (2019)), users are granted the right to request the removal of specific training data from models. To fulfill this requirement, machine unlearning have been extensively researched (Bourtoule et al. (2021); Yao et al. (2023)). However, applying these techniques to Transformers, which commonly involves a large number of parameters, poses a significant challenge in balancing effective unlearning with maintaining model fidelity (Liu et al. (2024c)).

Recent researches propose parameter-efficient unlearning techniques (Liu et al. (2024a); Pochinkov & Schoots (2024); Schoepf et al. (2024)), which identify the influence-critical parameters to govern the unlearning process. These methods assess the importance of parameters through different strategies, allowing selective updates to reduce computational overhead and improve unlearning efficiency. However, applying parameter-efficient unlearning to address the dilemma of the unlearning tasks in Transformers faces two major limitations. First, previous evaluation methods rely on heuristic or empirical strategies to identify parameters. For Transformer models with an immense number of parameters, identifying those specifically relevant to unlearning becomes inefficient. Additionally, existing methods (Pochinkov & Schoots (2024); Liu et al. (2023b); Shi et al. (2023)) assess importance of parameters by comparing performance (e.g., gradients or activations) on forget dataset and retain dataset may result in fine-grained selection process. Thus, previous unlearning methods overlook the intricate interactions between modules in Transformers. Transformers utilize parallel attention heads and hierarchical filters to perform computation and inference (Vaswani et al. (2017)). Consequently, attempting to identify critical parameters at a fine-grained level is often inaccurate, as this approach fails to capture the broader contextual relationships inherent in Transformers.

In this paper, we propose a **Module-aware Parameter-Efficient Unlearning** (`MAPE-Unlearn`) approach that targets influence-critical parameters at the module-level for Transformers. Specifically, `MAPE-Unlearn` formulates the unlearning objective through a pair of learnable masks applied to heads and filters. The derivation of this formulation ensures the effective removal of influences

and guides the identification of key modules. These masks are further refined by considering intra-layer interactions, and a warm-start greedy search algorithm is employed to efficiently optimize the process. Equipped with these module-aware masks, we integrate `MAPE-Unlearn` into various unlearning methods (e.g., second-order unlearning and gradient ascent). Additionally, we demonstrate that module-aware masks offer advantages in successive unlearning settings (Hu et al. (2024b); Liu et al. (2023a)) and against relearning attacks (Hu et al. (2024a); Łucki et al. (2024); Lynch et al. (2024)). While second-order unlearning introduces approximation errors, sparse updates using module-aware masks effectively limit these errors within selected modules, thus preserving overall performance in successive unlearning scenarios. Furthermore, by isolating parameter update regions, `MAPE-Unlearn` enhance robustness against relearning attacks. Our key contributions are summarized as follows:

- We introduce a new paradigm for identifying influence-critical parameters in Transformers, `MAPE-Unlearn`, which operates at the module-level. Our approach theoretically derives importance functions for selecting key modules using a pair of learnable masks. These module-aware masks can be seamlessly integrated into existing unlearning methods.

- We integrate `MAPE-Unlearn` into various unlearning methods and analyze the gains with module-aware masks. Extensive experiments on diverse tasks using different models demonstrate that the proposed method offers a superior trade-off between efficacy and fidelity.

- We evaluate the robustness of `MAPE-Unlearn` under complex unlearning settings and existing attacks. Empirical studies show that `MAPE-Unlearn` can handle a greater number of removal requests and is more resistant to relearning attacks compared to standard methods.

## 2 RELATED WORK

**Transformer Unlearning.** The concept of machine unlearning was first introduced by (Cao & Yang (2015)). Initially applied to simple model, machine unlearning has since been extended to Transformers (Jang et al. (2022); Eldan & Russinovich (2023); Yao et al. (2023; 2024); Chen et al. (2024); Jia et al. (2024); Gu et al. (2024)). (Jang et al. (2022)) proposed inverting the training objective on forgetting sequences and utilize straightforward gradient ascent (GA). As gradient ascent significantly degrades performance, (Liu et al. (2022); Yao et al. (2024)) introduced gradient difference (GD), which refines the objective function by employing gradient descent on in-distribution data to enhance robustness. Inspired by direct preference optimization (DPO) (Rafailov et al. (2024)), negative preference optimization (NPO) (Zhang et al. (2024a)) interprets the forget data as negative examples in preference alignment to deviate from the original model. Subsequently, (Jia et al. (2024)) provided a comprehensive overview of unlearning objectives and developed a second-order optimization unlearning approach. (Gu et al. (2024)) further investigated the effectiveness of second-order updates on Transformers. However, these methods primarily focus on updating all parameters, which is computationally expensive. In our work, we study the parameter-efficient methods to achieve effective unlearning on Transformers.

**Parameter-efficient Unlearning.** Parameter-efficient unlearning methods focus on identifying influence-critical parameters and updating only those to accelerate the unlearning process. Several strategies (Ma et al. (2022); Pochinkov & Schoots (2024); Shi et al. (2023); Liu et al. (2023b); Wu & Harandi (2024); Foster et al. (2024); Schoepf et al. (2024)) have been proposed to assess the importance of the parameters. Although these approaches may be applicable to Transformers, they are largely heuristic or empirical, which can lead to less reliable outcomes for unlearning tasks. Recently, (Liu et al. (2024a)) highlighted that unlearning can be effective when performed on a pruned model with a theoretical foundation. However, pruning focuses primarily on identifying parameters critical to maintain model performance, which does not align with the unlearning desiderata. Additionally, the focus on parameter ignores the complex intra-layer interactions within Transformers, which results in inaccurate identification of the parameters. Therefore, we specifically target modules within Transformers and derive an efficient strategy to identify influence-critical parameters.

## 3 PRELIMINARY

Machine unlearning is concerned with eradicating the influence of designated data instances from a trained model. Let $\mathcal{D} = \{x_i\}_{i=1}^{M}$ denote a training dataset containing $M$ data points, where each

$x_i$ denotes an individual data point. Given an initial model $\theta^*$ which was pre-trained on $\mathcal{D}$, the objective of unlearning is to effectively remove sensitive or compliance-related data points while maintaining overall performance. Specifically, the dataset $\mathcal{D}$ is partitioned into two disjoint subsets: **forget dataset** $\mathcal{D}_f$ and **retain dataset** $\mathcal{D}_r$, i.e., $\mathcal{D} = \mathcal{D}_f \cup \mathcal{D}_r$ and $\mathcal{D}_f \cap \mathcal{D}_r = \emptyset$. The forget dataset $\mathcal{D}_f$ consists of the data points we aim to remove from the model, while the retain dataset $\mathcal{D}_r$ includes the data points that should remain and may undergo further optimization. Given a loss function $\ell$ for the targeted task, the ultimate objective of unlearning can be framed as learning an optimal model:

$$\theta = \arg\min_\theta \mathcal{L}(\theta; \mathcal{D}_r) = \arg\min_\theta \sum_{x \in \mathcal{D}_r} \ell(\theta; x), \tag{1}$$

where $\mathcal{L}(\theta; \mathcal{D}_r)$ represents the total loss on the dataset $\mathcal{D}_r$ with $\theta$. We formalize this unlearning objective as Minimizing the Loss on the Retain dataset (MLR). The most straightforward solution to the optimization problem is retraining the model from scratch. However, retraining is often computationally expensive and time-consuming. As a efficient alternative, the Second-Order (SO) unlearning update method (Guo et al. (2020); Golatkar et al. (2020); Izzo et al. (2021); Warnecke et al. (2021); Liu et al. (2024b)) derives a generalized closed-form parameter modification directly from the original model based on MLR:

$$\theta \approx \theta^* + \mathbf{H}_{\theta^*}^{-1} \sum_{x \in \mathcal{D}_f} \nabla_\theta \ell(\theta^*; x), \tag{2}$$

where $\mathbf{H}_{\theta^*}^{-1}$ represents the inverse of the Hessian matrix $\nabla_\theta^2 \mathcal{L}(\theta^*; \mathcal{D}_r)$ evaluated at $\theta^*$ using the retain dataset $\mathcal{D}_r$. This approach builds upon the influence function (Koh & Liang (2017)), which provides a bounded approximation error to facilitate effective unlearning (Guo et al. (2020)). The implementation details of the second-order update are provided in Appendix A.1.

Following the generic formulation of unlearning, a new research direction has emerged. These optimization methods focus on fine-tuning the model with predefined objectives (Jia et al. (2024); Liu et al. (2024c)). Gradient Ascent (GA) (Jang et al. (2022)) performs reverse optimization on the loss over $\mathcal{D}_f$, while Negative Preference Optimization (NPO) (Zhang et al. (2024a)) targets maximizing the discrepancy between the original model and the unlearned model on $\mathcal{D}_f$. Despite differences in their optimization strategies, these methods share a common unlearning objective: Maximizing the Loss on the Forget dataset (MLF). This unlearning objective can be formalized as follows:

$$\arg\max_\theta \mathcal{L}(\theta; \mathcal{D}_f). \tag{3}$$

## 4 MODULE-AWARE MACHINE UNLEARNING

Inspired by the lottery hypothesis (Frankle & Carbin (2018)), recent research suggests that localizing functional regions within neural networks enhance their effectiveness for specific tasks (Zhang et al. (2024b)). However, for large models with high-dimensional parameter spaces, precisely identifying important parameters at a highly granular level is both inefficient and challenging. To this end, we propose `MAPE-Unlearn`, which introduces dual masks to locate influence-critical parameters within modules , tailored to different unlearning objectives as outlined in Section 4.1. Specifically, the multi-head attention mechanisms and feed-forward networks in LLMs serve as modules. Our method focuses on these modules (i.e., heads and filters). By selectively targeting these modules, `MAPE-Unlearn` is seamlessly integrated into various unlearning methods in Section 4.2.

### 4.1 MODULE-AWARE PARAMETER LOCALIZATION

While parameter-efficient methods involve identifying critical parameters, this process can be reformulated as the identification of an optimal binary mask. In this context, a mask value of $1$ indicates that the corresponding parameter should be updated, whereas a value of $0$ indicates it should remain frozen. Given that the number of modules is much smaller than the number of parameters (e.g., 37K vs. 110M in case of BERT-base), `MAPE-Unlearn` adopts a coarse-grained method to pinpoint influence-critical parameters in heads and filters. Thus, we formulate unlearning objective MLR (1) with a learnable pair of masks for the heads and filters as a constrained optimization problem (the parameter localization for MLF is similar; see Appendix B.1). To streamline the problem, we provide

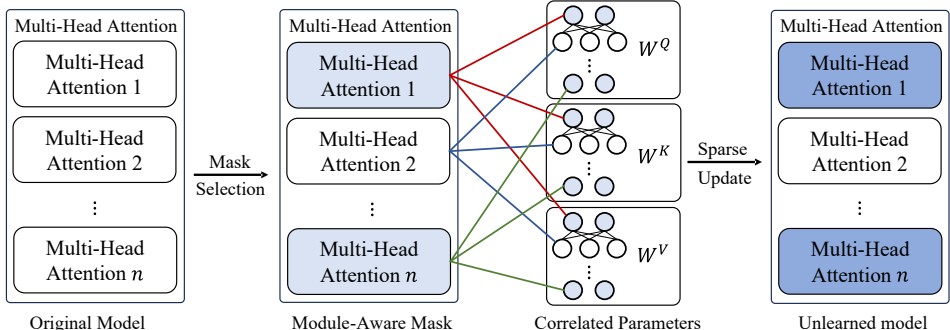

Figure 1: Illustration of our method applied to obtain important heads. Starting with the original model, key heads are identified highlighted in light blue. The different colored dashed lines (e.g., red, blue, green) represent the connections between heads and their correlated parameters. Last, we update the active parameters within heads highlighted in blue to represent unlearning process.

a general expression for the heads and filters by introducing the mask variables:

$$\mathbf{m}^* = \arg\min_{\mathbf{m}} \mathcal{L}(\mathbf{m}; \theta^*, \mathcal{D}_\mathrm{r}) \ \text{ s.t. } \frac{\|\mathbf{m}\|_0}{n} < 1 - \mathrm{S}, \tag{4}$$

where m is a binary vector representing the mask for heads/filters, and $\mathbf{m}_i$ corresponds to the $i$-th element. $\theta^*$ represents the original model, $n$ is the number of modules, and S denotes the sparsity (e.g., 90%) which determines the proportion of frozen modules. Since our focus is solely on the mask variables, we treat the parameters $\theta^*$ as fixed constants throughout. As a result, the total loss $\mathcal{L}(\theta^*; \mathcal{D}_\mathrm{r})$ can be mapped to $\mathcal{L}(\mathbf{m}; \theta^*, \mathcal{D}_\mathrm{r})$.

Since such constraint involves the $L_0$ norm of the mask m, which is non-differentiable, we approximate the optimization by assuming $\mathcal{L}$ is differentiable with respect to m. This allows us to apply a second-order Taylor series to $\mathcal{L}(\mathbf{m}; \theta^*, \mathcal{D}_\mathrm{r})$ around the mask variables $\mathbb{1}$:

$$\mathcal{L}(\mathbf{m}; \theta^*, \mathcal{D}_\mathrm{r}) \approx \mathcal{L}(\mathbb{1}; \theta^*, \mathcal{D}_\mathrm{r}) - (\mathbb{1}-\mathbf{m})^\mathrm{T} \nabla_\mathbf{m} \mathcal{L}(\mathbb{1}; \theta^*, \mathcal{D}_\mathrm{r}) + \frac{1}{2}(\mathbb{1}-\mathbf{m})^\mathrm{T} \nabla_\mathbf{m}^2 \mathcal{L}(\mathbb{1}; \theta^*, \mathcal{D}_\mathrm{r})(\mathbb{1}-\mathbf{m}). \tag{5}$$

As the original model $\theta^*$ is assumed to have converged to a local minimum of $\nabla_\mathbf{m}\mathcal{L}(\mathbb{1}; \theta^*, \mathcal{D})$, we can take $\nabla_\mathbf{m}\mathcal{L}(\mathbb{1}; \theta^*, \mathcal{D}) = 0$ (LeCun et al. (1989)). Incorporating this assumption, we simplify gradient term in the Taylor series approximation. Specifically, $\nabla_\mathbf{m}\mathcal{L}(\mathbb{1}; \theta^*, \mathcal{D}_\mathrm{r}) = \nabla_\mathbf{m}\mathcal{L}(\mathbb{1}; \theta^*, \mathcal{D}) - \sum_{x \in \mathcal{D}_\mathrm{f}} \nabla_\mathbf{m}\ell(\mathbb{1}; \theta^*, x) = -\sum_{x \in \mathcal{D}_\mathrm{f}} \nabla_\mathbf{m}\ell(\mathbb{1}; \theta^*, x)$. As $\mathcal{L}(\mathbb{1}; \theta^*, \mathcal{D}_\mathrm{r})$ is constant, we can reformulate the unlearning objective in terms of the mask variables:

$$\mathbf{m}^* \approx \arg\min_{\mathbf{m}}(\mathbb{1} - \mathbf{m})^\mathrm{T} \sum_{x \in \mathcal{D}_\mathrm{f}} \nabla_\mathbf{m}\ell(\mathbb{1}; \theta^*, x) + \frac{1}{2}(\mathbb{1} - \mathbf{m})^\mathrm{T} \nabla_\mathbf{m}^2 \mathcal{L}(\mathbb{1}; \theta^*, \mathcal{D}_\mathrm{r})(\mathbb{1} - \mathbf{m}). \tag{6}$$

Thus, the optimization problem depends on the two factors: the gradient with respect to the forget dataset $\mathcal{D}_\mathrm{f}$ (i.e., $\sum_{x \in \mathcal{D}_\mathrm{f}} \nabla_\mathbf{m}\ell(\mathbb{1}; \theta^*, x)$) and the Hessian matrix with respect to the retain dataset $\mathcal{D}_\mathrm{r}$ (i.e., $\nabla_\mathbf{m}^2 \mathcal{L}(\mathbb{1}; \theta^*, \mathcal{D}_\mathrm{r})$). These components collectively reflect the effectiveness of influence removal (Guo et al. (2020)). Since forming the Hessian matrix directly is computationally prohibitive, we approximate it using the empirical **diagonal Fisher Information Matrix (FIM)** of the mask variables (details provided in Appendix A.1). This leads to a simplified form of optimization objective:

$$\mathbf{m}^* \approx \arg\min_{\mathbf{m}}(\mathbb{1} - \mathbf{m})^\mathrm{T} \sum_{x \in \mathcal{D}_\mathrm{f}} \nabla_\mathbf{m}\ell(\mathbb{1}; \theta^*, x) + \frac{1}{2}(\mathbb{1} - \mathbf{m})^2 \widehat{\mathcal{I}}(\mathbb{1}; \theta^*, \mathcal{D}_\mathrm{r}), \tag{7}$$

where $\widehat{\mathcal{I}}(\mathbb{1}; \theta^*, \mathcal{D}_\mathrm{r})$ represents the diagonal FIM evaluate at $\theta^*$ using $\mathcal{D}_\mathrm{r}$. Given that the mask variable can only take binary values (0 or 1), we transform the optimization problem into a discrete mask selection problem over modules:

$$\mathbf{m}^* \approx \arg\min_{\mathbf{m}} \sum_i \left[ (1 - \mathbf{m}_i)\big[ \sum_{x \in \mathcal{D}_\mathrm{f}} \nabla_\mathbf{m}\ell(\mathbb{1}; \theta^*, x) \big]_i + \frac{1}{2}(1 - \mathbf{m}_i)^2 \big[\widehat{\mathcal{I}}(\mathbb{1}; \theta^*, \mathcal{D}_\mathrm{r})\big]_i \right]. \tag{8}$$

Therefore, we propose importance scores to identify influence-critical modules. Each module can be assessed based on the sum of its corresponding gradient and half of the diagonal FIM element.

Modules with higher scores will be prioritized for selection. Additionally, to better understand the influence of off-diagonal elements on mask selection for each layer, we replace the diagonal FIM with the **block diagonal FIM**, where each block is associated with a layer. Thus, Equation (7) decomposes into *layer-wise* optimization problems:

$$\mathrm{m}_l^* \approx \arg\min_{\mathrm{m}_l}(\mathbb{1} - \mathrm{m}_l)\big[ \sum_{x \in \mathcal{D}_{\mathrm{f}}} \nabla_{\mathrm{m}}\ell(\mathbb{1}; \theta^*, x)\big]_l + \frac{1}{2}(\mathbb{1} - \mathrm{m}_l)^2\big[\widehat{\mathcal{I}}(\mathbb{1}; \theta^*, \mathcal{D}_{\mathrm{r}})\big]_l, \qquad (9)$$

where $l$ represents the layer being optimized, and $\mathrm{m}_l$ denotes the mask variable in the $l$-th layer. This optimization problem can be efficiently solved using a greedy search with warm start (Kwon et al. (2022)), i.e., by initializing the mask variable $\mathrm{m}_l$ based on Equation (8). In this process, we iteratively swap each unselected module having the highest importance score with a selected one in the current mask to further optimize Equation (9), yielding an approximate solution after one round of swapping. Consequently, the rearranged mask variables capture the impact of intra-layer interactions, enabling precise localization of the parameters within the model's modules. Additionally, our approach can be integrated with other methods for identifying influence-critical parameters, offering enhanced flexibility. Detailed information can be found in Appendix B.2.

### 4.2 APPLICATIONS OF MAPE-UNLEARN

By identifying influence-critical parameters within modules, `MAPE-Unlearn` facilitates efficient integration with widely adopted unlearning methods. Below, we examine the benefits of module-aware machine unlearning across various unlearning approaches.

**Module-Aware Second-Order Updates.** Following the insights of `MAPE-Unlearn`, we formalize Module-aware Parameter-Efficient Second-Order unlearning (MAPE-SO) by introducing sparse mask variables linked to the outputs of modules:

$$\theta \approx \theta^* + \mathrm{m} \circ \Big[\mathbf{H}_{\theta^*}^{-1} \sum_{x \in \mathcal{D}_{\mathrm{f}}} \nabla_\theta \ell(\theta^*; x)\Big], \qquad (10)$$

where m are module-aware mask variables derived from the MLR objective (1), and $\circ$ denotes the Hadamard product. In practice, the computation of the Hessian matrix and gradients is restricted to the parameters associated with a module-aware mask value of 1. Notably, Equation (2) can be interpreted by setting all mask variables to 1.

We now provide an intuitive analysis of the benefits of MAPE-SO. First, by incorporating sparsity through module-aware masks, MAPE-SO significantly reduces the number of parameters required for the expensive computation of the Hessian matrix. This leads to lower computational complexity, making the method more scalable and efficient when applied to large-scale models. Second, MAPE-SO offers a more tightly bounded approximation error compared to standard method. The approximation error is reduced by a factor that is directly proportional to the sparsity introduced by the mask variables. This ensures that the unlearning process remains highly accurate while avoiding unnecessary parameter updates. Furthermore, by restricting the influence-critical parameters within the modules, MAPE-SO provides fine-grained control over the error bounds.

We also consider **successive unlearning**, a practical scenario in which data owners request the removal of data from the model at intervals (e.g., in machine learning as a service (MLaaS) (Hu et al. (2024b))). For each unlearning request, the model progressively applies unlearning algorithms, building on the state from the previous unlearning cycle (Guo et al. (2020); Gu et al. (2024)). However, since SO inherently diverges from the Taylor series approximation, small errors arise during each approximation. These errors accumulate with successive updates, causing the model to diverge from its original state. As a result, with an increasing number of unlearning requests, the gap between the original and updated models widens, leading to a gradual decline in performance.

Table 1: Accuracy results of standard SO under varying removal requests on the MNLI dataset using the BERT-base model.

| Requests | 1 | 4 | 8 | 10 |
|---|---|---|---|---|
| Test Acc. | 84.34% | 83.86% | 83.60% | 83.46% |
| Remain Acc. | 94.33% | 94.18% | 94.05% | 93.86% |

Once the number of requests exceeds a certain threshold, retraining the model from scratch becomes necessary to restore performance (detailed in Table 1). Fortunately, MAPE-SO mitigates this issue by allowing a greater number of removal requests before retraining becomes essential (as shown in

Figure 3). This improvement stems from selectively adjusting only the modules directly related to the removed data. By confining cumulative errors to a subset of parameters, MAPE-SO reduces the overall impact on model performance. Consequently, the model remains robust even after multiple unlearning operations, delaying the need for costly retraining. Additionally, we explore an alternative successive unlearning scenario, as described in (Liu et al. (2023a)), detailed in Appendix A.2.

**Module-Aware Fine-Tuning Based Unlearning.** Mainstream unlearning methods based on fine-tuning typically aim to increase the loss on the forget data, motivating the use of the MLF objective (3) to identify key parameters and optimize the unlearning process. Building on this, these methods can be reformulated using `MAPE-Unlearn` (e.g., MAPE-GA and MAPE-NPO).

In our approach, we use module-aware masks to implement unlearning via sparse updates, targeting key parameters that align with the unlearning objective to fulfill the desired outcomes. This method is inspired by pruning strategies (Liu et al. (2024a); Pochinkov & Schoots (2024)), where the model is adjusted iteratively using sparse updates instead of directly removing parameters, thereby preserving overall model performance. Since unlearning typically involves a performance trade-off, freezing non-critical parameters can better maintain model performance compared to updating all parameters.

However, recent work has demonstrated that machine unlearning can be trivially compromised by **relearning attacks** (Hu et al. (2024a); Łucki et al. (2024); Doshi & Stickland (2024)). These attacks use unrelated or orthogonal data to recover previously unlearned knowledge with just a few fine-tuning steps. Notably, parameter-efficient tuning methods, such as LoRA (Hu et al. (2021)), are particularly susceptible to these attacks compared to full parameter fine-tuning (Hu et al. (2024a)). We hypothesize that this vulnerability stems from the introduced parameters, which may encode residual information from the forget data during the unlearning process. In contrast, `MAPE-Unlearn` employs sparse update with module-aware masks, which restricts the unlearning scope and disrupts the pathways for knowledge recovery. Furthermore, by keeping most parameters frozen, the model retains its ability to handle normal data, thereby increasing the difficulty of selecting effective data for attacks. Consequently, `MAPE-Unlearn` demonstrates enhanced robustness against relearning attacks.

## 5 EXPERIMENTS

### 5.1 EXPERIMENT SETUPS

**Models and Datasets.** Our empirical analysis is conducted on three well-established unlearning tasks. **1) Traditional Task.** We consider three GLUE benchmarks (MNLI, QQP, SST-2) (Wang et al. (2018)) for the classification task and two SQuAD benchmarks (SQuAD v1.1 and SQuAD v2.0) (Rajpurkar (2016)) for the question-answering task. For each benchmark, we randomly select 128 samples as the forget dataset $\mathcal{D}_f$, while all orthogonal samples serves as the retain dataset $\mathcal{D}_r$. These benchmark are evaluated on two pretrained models: BERT-base (Devlin et al. (2018)) and RoBERTa-large (Liu et al. (2019)). **2) Task of Fictitious Unlearning (TOFU).** The unlearning scenarios of TOFU (Maini et al. (2024)) are categorized into three types: Forget01, Forget05, and Forget10, corresponding to 1%, 5%, and 10% of the training dataset, respectively. We use the LLama2-7b-chat model (Touvron et al. (2023)) provided by the TOFU benchmark to evaluate three scenarios. **3) Hazardous Knowledge Removal Task.** This task is evaluated using the WMDP benchmark (Li et al. (2024)), which measures hazardous capabilities across three domains (biology, cybersecurity, and chemistry). We unlearn the bio-attack corpus and cyber-attack corpus using the Zephyr-7B-beta model (Tunstall et al. (2023)). More details are deferred to Appendix D.1.

**Unlearning methods.** We demonstrate the effectiveness of `MAPE-Unlearn` by comparing it with several unlearning baselines: Gradient Ascent (GA) (Jang et al. (2022)), Gradient Difference (GD) (Liu et al. (2022)), Negative Preference Optimization (NPO) (Zhang et al. (2024a)). We also consider Direct Preference Optimization (DPO) (Rafailov et al. (2024)), which uses a reject-based answer 'I don't know' on alignment loss on the TOFU task. In our approach, these methods identify important parameters with MLF as the unlearning objective. For traditional tasks, Second-Order (SO) serves as a baseline, while the unlearning objective MLR is used to identify important parameters in our method. We consider saliency-based unlearning with a large learning rate (SURE) (Zhang et al. (2024c)) as baseline, which also operates at the module-level and directly uses the gradient norm of the forget dataset. Sparsity-Aware Unlearning (SA) (Liu et al. (2024a)) is another method for sparse unlearning, which fine-tunes the retain dataset with a sparsity penalty ($\gamma = 5e - 5$). Meanwhile,

Retraining from scratch (RT) serves as the gold standard for traditional and TOFU tasks, where the model is fine-tuned on the retain dataset. Detailed hyperparameters are provided in Appendix D.2.

**Evaluation metrics.** We evaluate the unlearning performance based on two key aspects: unlearning **efficacy** and model **fidelity**. For the **Traditional Tasks**, the evaluation metrics differ depending on the specific tasks. Accuracy is used for the classification task, while F1 scores are reported for question-answering task. Unlearning accuracy (or F1 score) directly reflects the effectiveness of the unlearning algorithm. We also consider membership inference attacks (MIA) to assess the vulnerability of the model to attacks, which helps evaluate unlearning efficacy. In practice, we use a confidence-based MIA predictor to gauge the likelihood of a successful attack (Liu et al. (2024a); Song et al. (2019)). Model fidelity is measured by evaluating both retain accuracy and test accuracy, which assess the preservation of model performance and its generalization ability after unlearning. For the **TOFU task**, we use normalized probability, ROUGE scores, and truth ratio (TR) (Maini et al. (2024)) on the forget dataset to preliminarily assess unlearning efficacy, where TR represents the preference probability between incorrect and correct answers. These metrics on retain, real authors and world facts datasets are used to evaluate model fidelity, which are then aggregated to represent model utility (MU). Furthermore, unlearning efficacy can be further measured by forget quality (FQ), which quantifies the performance difference between unlearned model and retrained model. Specifically, FQ and MU represent comprehensive considerations of unlearning efficacy and fidelity, respectively. For the **Hazardous Knowledge Removal Task**, we use accuracy on WMDP-Bio and WMDP-Cyber datasets to measure unlearning efficacy, while zero-shot accuracy on the MMLU dataset (Hendrycks et al. (2020)) is used to measure model fidelity.

## 5.2 RESULTS ON TRADITIONAL TASKS

We present the experimental results on traditional tasks using the SQuAD v2.0 dataset as a case study. Detailed results for additional datasets are provided in Appendix D.4. In what follows, we compare different unlearning methods and conduct an in-depth analysis of our approach.

Table 2: Overall results of unlearning performance using different unlearning methods under two fine-tuned models on SQuAD v2.0.

| | BERT | | | | | RoBERTa | | | |
|---|---|---|---|---|---|---|---|---|---|
| | **Efficacy** | | **Fidelity** | | | **Efficacy** | | **Fidelity** | |
| **Method** | Unlearn F1. | MIA | Retain F1. ↑ | Test F1. ↑ | **Method** | Unlearn F1. | MIA | Retain F1. ↑ | Test F1. ↑ |
| RT | 73.77 | 0.6484 | 98.72 | 75.77 | RT | 87.03 | 0.7734 | 98.42 | 86.58 |
| SA | 79.16 | 0.6797 | **96.03** | 72.65 | SA | 84.05 | 0.7343 | 93.21 | 80.82 |
| SO | 78.03 | 0.6797 | 93.66 | 73.33 | SO | 87.70 | **0.7188** | 94.68 | 85.22 |
| SURE-SO | 78.09 | 0.7109 | 92.62 | 71.69 | SURE-SO | 87.34 | 0.7344 | 94.72 | 85.45 |
| MAPE-SO | **77.40** | **0.6563** | **93.90** | 73.57 | MAPE-SO | **87.34** | **0.7188** | **94.76** | **85.50** |
| GA | 76.37 | 0.7500 | 86.61 | 72.24 | GA | 88.58 | 0.7734 | **94.87** | 85.54 |
| SURE-GA | **75.33** | 0.7109 | 88.93 | 74.61 | SURE-GA | 88.39 | **0.7343** | 94.72 | 85.44 |
| MAPE-GA | **75.33** | **0.7031** | **89.68** | **74.81** | MAPE-GA | 88.39 | 0.7343 | **94.87** | **85.55** |
| GD | 78.77 | 0.7109 | 94.36 | 74.25 | GD | 88.82 | 0.7891 | 94.86 | 85.56 |
| SURE-GD | 78.10 | 0.6953 | 94.45 | 76.28 | SURE-GD | 88.56 | 0.7813 | 94.78 | 85.72 |
| MAPE-GD | **74.56** | **0.6875** | **94.60** | **76.59** | MAPE-GD | 88.56 | **0.7500** | **94.91** | **85.80** |
| NPO | 77.71 | 0.7266 | **93.96** | 76.17 | NPO | 87.88 | 0.7500 | 94.78 | 85.42 |
| SURE-NPO | 75.59 | 0.6953 | 93.10 | 76.43 | SURE-NPO | 87.10 | 0.7500 | 94.82 | 85.79 |
| MAPE-NPO | **74.81** | **0.6719** | 93.12 | **76.50** | MAPE-NPO | **87.10** | **0.7344** | **94.88** | **85.95** |

**Module-Aware sparse unlearning is effective.** Table 2 compares the unlearning performance of various methods across two models. Our experiments demonstrate that 90% sparsity suffices for effective unlearning; thus, we focus on this regime for baseline comparisons. While the sparse weight-based method (SA) achieves strong unlearning efficacy, it significantly compromises model fidelity. However, its performance on primary tasks lags behind mainstream methods (SO, GA, GD, and NPO). In contrast,

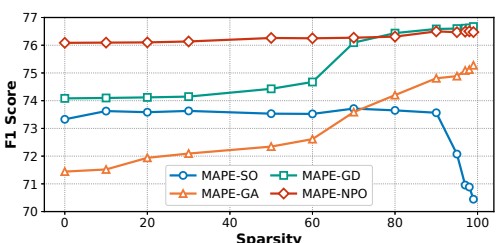

Figure 2: F1 scores of unlearning methods on BERT-base at different sparsity levels.

`MAPE-Unlearn` consistently outperforms counterparts in unlearning efficacy, evidenced by higher

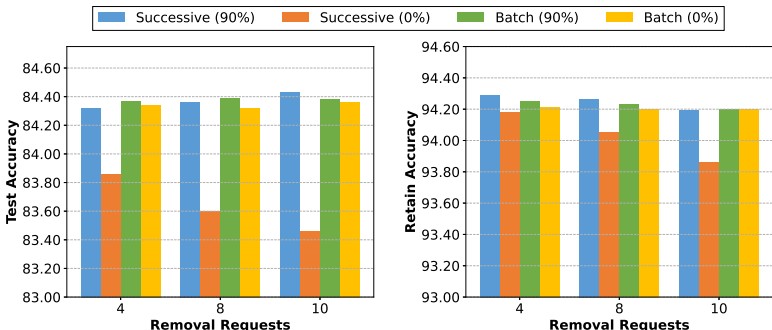

Figure 3: Results on BERT-base under different unlearning scenarios with varying removal requests. 'Successive' refers to successive unlearning scenarios, while 'batch' refers to batch unlearning scenarios. The numbers following each unlearning scenario indicate the corresponding sparsity.

F1 scores and lower MIA success rates on the forget set. Moreover, `MAPE-Unlearn` maintains strong model fidelity, achieving utility performance competitive with baselines. Notably, gradient-based updates on the forget dataset alone (SURE-based methods) also enable effective module-aware unlearning. Among these, MAPE-GD and MAPE-NPO achieve an optimal trade-off between unlearning efficacy and fidelity preservation. These results demonstrate that module-aware sparse updates offer an effective approach to machine unlearning.

**A sparsity of 90% is sufficient for effective unlearning.** We explore the effectiveness of various sparsity strategies in facilitating unlearning. Figure 2 shows the relationship between test score and sparsity, while maintaining comparable unlearning efficacy. Unlearning methods based on fine-tuning show a continuous improvement in utility as sparsity increases. However, when sparsity surpasses 90%, SO experiences a sharp decline in test scores with optimal utility observed at a sparsity level of 70%. Additionally, we also delve into the functional regions responsible for unlearning within models, but find no single network layer that stands out as particularly crucial for unlearning. This suggests that the effectiveness of unlearning may be task-specific, resisting any fixed modular or parametric approach. Overall, our findings highlight that a 90% sparsity strategy offers a sufficient balance between efficacy and fidelity.

**Successive unlearning benefits from MAPE-SO.** We further explore the potential of module-aware masks in successive unlearning scenarios on SO, as depicted in Figure 3. Our results show that sparse updates with module-aware offer significant benefits over full parameters updates. When all parameters are updated, model fidelity is overly impacted. However, applying sparse updates with 90% sparsity preserves high utility, even after 10 removal requests. This suggests that module-aware masks can support a higher volume of removal requests before retraining becomes necessary. These results highlight the potential of module-aware masks to enhance the robustness of unlearning.

## 5.3 Results on TOFU Task

We provide the experimental results for the TOFU task, using Forget05 as a representative example. Additional results for other types are available in Appendix D.5. The subsequent analysis focuses specifically on unlearning methods based on fine-tuning. In Table 3, we showcase the unlearning performance of `MAPE-Unlearn` and its baselines for the TOFU task. As observed, SA is not an effective unlearning method for this task. In contrast, the incorporation of `MAPE-Unlearn` consistently enhances across most unlearning method categories (with the exception of GA). We attribute this improvement to the relatively small amount of unlearning required. Our method shows exceptional performance on Forget10 (see Table 9). Furthermore, we observe that PO-based methods (i.e., DPO and NPO) are particularly effective, delivering high forget quality and model utility.

**MAPE-Unlearn is resistant to relearning attacks.** We further evaluate the robustness of `MAPE-Unlearn` against relearning attacks by randomly selecting 20% of the dataset samples as the relearning dataset. Since forget quality is obtained through the Kolmogorov-Smirnov test, achieving higher values is highly sensitive to the hyperparameters, especially for GA and GD. Thus, we experimented with various hyperparameters to achieve the reasonable forget quality. Figure 4 illustrates the trend of forget quality as the number of relearning epochs increases. Our method withstand a longer attack duration on GA and GD. For DPO, full-parameter updates can effectively

Table 3: Performance overview of various unlearning methods on Forget05 unlearning settings for TOFU under LLama2-7b-chat model.

| Method | Efficacy | | | | Fidelity | | | | | | | | | |
| --- | --- | --- | --- | --- | --- | --- | --- | --- | --- | --- | --- | --- | --- | --- |
| | Forget Set | | | FQ ↑ | Real Authors | | | World Facts | | | Retain Set | | | MU ↑ |
| | Rouge | Prob. | TR | | Rouge ↑ | Prob. ↑ | TR ↑ | Rouge ↑ | Prob. ↑ | TR ↑ | Rouge ↑ | Prob. ↑ | TR ↑ | |
| RT | 0.39 | 0.15 | 0.67 | 1.0 | 0.96 | 0.42 | 0.55 | 0.90 | 0.40 | 0.53 | 0.98 | 0.99 | 0.46 | 0.62 |
| SA | 0.97 | 0.99 | 0.51 | 3.43e-16 | 0.94 | 0.45 | 0.58 | 0.87 | 0.42 | 0.55 | 0.98 | 0.99 | 0.48 | 0.62 |
| GA | **0.36** | **0.06** | **0.56** | 1.39e-6 | **0.84** | **0.29** | 0.39 | 0.87 | 0.35 | 0.48 | 0.41 | 0.18 | 0.49 | **0.38** |
| SURE-GA | **0.36** | 0.01 | 0.55 | **7.54e-5** | 0.80 | 0.32 | **0.44** | **0.88** | **0.36** | **0.50** | **0.42** | 0.08 | 0.49 | 0.30 |
| MAPE-GA | **0.36** | 0.04 | **0.56** | 1.83e-5 | 0.84 | 0.26 | 0.37 | **0.88** | **0.36** | 0.48 | 0.41 | **0.19** | 0.49 | **0.38** |
| GD | **0.39** | 0.11 | 0.52 | 2.43e-10 | 0.85 | 0.37 | 0.51 | 0.87 | 0.38 | 0.52 | 0.52 | 0.61 | 0.48 | **0.52** |
| SURE-GD | 0.37 | 0.02 | 0.52 | 4.61e-10 | 0.81 | **0.39** | **0.53** | 0.86 | 0.37 | 0.53 | 0.52 | 0.30 | 0.50 | 0.48 |
| MAPE-GD | 0.37 | 0.08 | 0.52 | **1.87e-9** | **0.86** | 0.36 | 0.51 | **0.89** | 0.39 | **0.54** | 0.52 | 0.46 | **0.50** | **0.52** |
| DPO | 0.20 | **0.14** | 0.72 | 0.27 | 0.60 | 0.34 | 0.44 | 0.73 | 0.37 | 0.50 | 0.37 | 0.47 | 0.37 | 0.44 |
| SURE-DPO | 0.25 | 0.05 | **0.68** | 0.39 | 0.43 | 0.35 | 0.46 | 0.70 | 0.38 | 0.54 | 0.41 | 0.47 | **0.40** | 0.44 |
| MAPE-DPO | **0.26** | 0.06 | **0.68** | **0.47** | **0.63** | **0.43** | **0.55** | **0.76** | **0.43** | **0.58** | **0.42** | 0.47 | 0.39 | **0.50** |
| NPO | 0.28 | 0.05 | **0.75** | 1.43e-3 | 0.89 | **0.42** | **0.55** | 0.82 | 0.41 | **0.55** | 0.43 | 0.33 | 0.34 | 0.47 |
| SURE-NPO | 0.29 | 0.04 | 0.72 | 0.02 | 0.90 | 0.39 | 0.52 | 0.86 | 0.41 | 0.54 | 0.43 | 0.41 | **0.39** | **0.49** |
| MAPE-NPO | **0.30** | **0.08** | 0.72 | **0.27** | **0.94** | 0.38 | 0.50 | **0.89** | 0.41 | 0.51 | **0.48** | **0.42** | **0.39** | **0.49** |

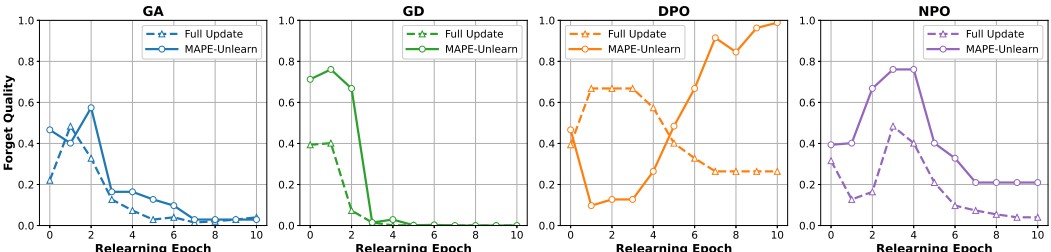

Figure 4: Forget quality for different unlearning methods with varying relearning epochs on TOFU Forget05. The dashed line represents full parameter updates, while the solid line represents `MAPE-Unlearn` with 90% sparsity.

defend against the relearning attack. Although our method experiences an initial decline, it ultimately achieves a high FQ. This can be attributed to the fact that, during the relearning process, the model relearns the correct examples (rather than 'I don't know'). In the case of NPO, our method successfully defends against the relearning attack, while full-parameter updates fail to do so. Overall, our method demonstrates superior robustness compared to full-parameter updates.

## 5.4 RESULTS ON HAZARDOUS KNOWLEDGE REMOVAL TASK

We present the unlearning performance in the hazardous knowledge removal task on WMDP. Given that retraining large language models to obtain a harmless model is impractical, we use the 'Original' model as a benchmark to assess both unlearning efficacy and model fidelity. As detailed in Appendix D.6, `MAPE-Unlearn` demonstrates strong competitiveness compared to the baselines, indicating its effectiveness in hazardous knowledge removal. These results are consistent with those observed in the other two tasks. This suggests that identifying harmful knowledge directly within the parameters can also serve as an effective approach for enabling unlearning. Overall, these findings further reinforce the potential of `MAPE-Unlearn` to address the challenge of removing hazardous knowledge.

## 6 CONCLUSION

In this work, we propose Module-Aware Parameter-Efficient unlearning (`MAPE-Unlearn`) for Transformers. `MAPE-Unlearn` develops an optimal masking strategy to identify influence-critical parameters within modules. By selectively targeting these key parameters, `MAPE-Unlearn` can be integrated into various unlearning methods to demonstrate its effectiveness. We further examine the advantages of our approach across second-order unlearning and fine-tuning-based unlearning. Extensive experiments conducted on various models and tasks show that our method with 90% sparsity outperforms existing approaches. Additionally, empirical studies on successive unlearning scenarios and relearning attacks highlight the robustness of our method.

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

# A    ADDITIONAL DETAILS OF SECOND-ORDER UNLEARNING UPDATE

## A.1    SIMPLIFIED SECOND-ORDER UNLEARNING UPDATE

Second-order unlearning involves the inverse Hessian computation, which is highly sensitive to parameters. Given the large number of parameters in large-scale models, this unlearning method cannot be applied directly. A common practice to approximate it is to use empirical FIM (Peste et al. (2021); Liu et al. (2024a); Gu et al. (2024)). Additionally, studies (Amari et al. (2019)) have shown that the off-diagonal elements of the FIM tend to be much smaller than the diagonal elements, usually by a factor $\frac{1}{\sqrt{n}}$, where $n$ represents the dimension of the FIM. This insight highlights the effectiveness of using a diagonal approximation, particularly in large models with vast parameter counts (Hwang (2024)). As a result, we further adopt the empirical diagonal FIM $\widehat{\mathcal{I}}$ to approximate the Hessian matrix:

$$\widehat{\mathcal{I}}(\theta;\mathcal{D}) = \frac{1}{|\mathcal{D}|}\sum_{x\in\mathcal{D}}\nabla\ell(\theta;x)^2. \tag{11}$$

The storage of the diagonal FIM requires only $\mathcal{O}(d)$ space, and the inverse operation takes only $\mathcal{O}(d)$ time, where $d$ denotes the number of model parameters. This makes second-order unlearning method straightforward and efficient to implement.

## A.2    ANOTHER SUCCESSIVE UNLEARNING SCENARIO

(Liu et al. (2023a)) introduced a successive unlearning scenario leveraging second-order updates, requiring the retention of data information (e.g., gradients and FIM) for efficient unlearning on the original model. Unlike previous methods, this approach directly employs a Taylor expansion around the original model parameters. Specifically, at the $t$-th unlearning request, MAPE-SO update at timestamp $t$ can be expressed as follows:

$$\mathrm{m}^t \circ \left\{ \left[ \frac{|\mathcal{D}_{\mathrm{r}}^{t-1}| \cdot \widehat{\mathcal{I}}(\theta^*;\mathcal{D}_{\mathrm{r}}^{t-1}) - \widehat{\mathcal{I}}(\theta^*;x^t)}{|\mathcal{D}_{\mathrm{r}}^{t-1}-1|} \right]^{-1} \left[ \sum_{x\in\mathcal{D}_{\mathrm{f}}^{t-1}} \nabla_\theta\ell(\theta^*;x) + \nabla_\theta\ell(\theta^*;x^t) \right] \right\}, \tag{12}$$

where $\mathcal{D}_{\mathrm{f}}^{t-1}$ represents the data points that have already been removed at timestamp $t-1$, $\mathcal{D}_{\mathrm{r}}^{t-1}$ denotes the retain dataset at timestamp $t-1$. In practice, rather than storing these data points directly, we retain the gradients or FIM associated with the data in memory. With each new unlearning request, these data information are updated accordingly. Furthermore, considering the proportion of the forget dataset is negligible, the mask selection process can be accelerated. As a result, in the mask selection Equation (8), the term $\sum_{x\in\mathcal{D}_{\mathrm{f}}}\nabla_m\ell(\mathbb{1};x)$ can be omitted, and the term $\widehat{\mathcal{I}}(\mathbb{1};\mathcal{D}_{\mathrm{r}})$ can be approximated by $\widehat{\mathcal{I}}(\mathbb{1};\mathcal{D})$, resulting in the following simplification:

$$\mathrm{m}^* \approx \arg\min_{\mathrm{m}} \sum_i (\mathbb{1}-\mathrm{m}_i)^2\widehat{\mathcal{I}}(\mathbb{1};\mathcal{D})_i, \tag{13}$$

This simplification allows the mask to be pre-computed in the pre-unlearning phase, improving efficiency. However, it does not fully account for the influence of the data slated for deletion. To address this, mask variables are refined using Equation (9) for more targeted adjustment. In this setting, unlearning is achieved via a single-step second-order update on the original model. The effectiveness of module-aware masks lies in the ability of `MAPE-Unlearn` to handle general second-order unlearning, offering tighter approximation error bounds for more precise and efficient data removal.

## B  EXTENSIONS TO OUR METHOD

### B.1  IDENTIFY KEY MODULES WITH THE OBJECTIVE OF MAXIMIZING THE LOSS ON THE FORGET DATASET

The task of identifying key modules under the objective of maximizing the loss on the forget dataset shares conceptual similarities with identifying key modules under the objective of minimizing the loss on the retained dataset. By exploiting this similarity, we can efficiently adopt a comparable approach to locate influence-critical parameters. To this end, we introduce a learnable pair of binary masks, denoted as m, applied to the heads and filters in Transformers. The mask variable m specifies which parameters are updated during the optimization process. The resulting optimization problem can be formulated as:

$$\mathrm{m}^* = \arg\max_{\mathrm{m}} \mathcal{L}(\mathrm{m}; \theta^*, \mathcal{D}_{\mathrm{f}}) \quad \mathrm{s.t.} \frac{\|\mathrm{m}\|_0}{n} < 1 - \mathrm{S}, \tag{14}$$

where $|\mathrm{m}|$ is the number of mask variables, $\theta^*$ represents the original model, and S denotes the sparsity constraint. We then approximate it using the second-order Taylor series around the mask variables $\mathbb{1}$:

$$\mathcal{L}(\mathrm{m}; \theta^*, \mathcal{D}_{\mathrm{f}}) \approx \mathcal{L}(\mathbb{1}; \theta^*, \mathcal{D}_{\mathrm{f}}) - (\mathbb{1}-\mathrm{m})^{\mathrm{T}} \nabla_{\mathrm{m}} \mathcal{L}(\mathbb{1}; \theta^*, \mathcal{D}_{\mathrm{f}}) + \frac{1}{2}(\mathbb{1}-\mathrm{m})^{\mathrm{T}} \nabla_{\mathrm{m}}^2 \mathcal{L}(\mathbb{1}; \theta^*, \mathcal{D}_{\mathrm{f}})(\mathbb{1}-\mathrm{m}). \tag{15}$$

To further streamline the computation, we use the diagonal Fisher Information Matrix (FIM) to approximate the Hessian matrix, which simplifies the second-order term. Omitting constant terms, the optimization objective becomes:

$$\mathrm{m}^* \approx \arg\max_{\mathrm{m}} (\mathbb{1} - \mathrm{m})^{\mathrm{T}} \sum_{x \in \mathcal{D}_{\mathrm{r}}} \nabla_{\mathrm{m}} \ell(\mathbb{1}; \theta^*, x) + \frac{1}{2}(\mathbb{1} - \mathrm{m})^2 \widehat{\mathcal{I}}(\mathbb{1}; \theta^*, \mathcal{D}_{\mathrm{f}}). \tag{16}$$

Since the mask can only take binary values (0 or 1), we derive the importance evaluation function as:

$$\mathrm{m}^* \approx \arg\max_{\mathrm{m}} \sum_i \left[ (1 - \mathrm{m}_i) \big[ \sum_{x \in \mathcal{D}_{\mathrm{r}}} \nabla_{\mathrm{m}} \ell(\mathbb{1}; \theta^*, x) \big]_i + \frac{1}{2}(1 - \mathrm{m}_i)^2 \big[ \widehat{\mathcal{I}}(\mathbb{1}; \theta^*, \mathcal{D}_{\mathrm{f}}) \big]_i \right]. \tag{17}$$

To further refine the optimization process, we employ the block diagonal FIM, which provides a more localized approximation for each layer in the model. The layer-wise optimization can be expressed as:

$$\mathrm{m}_l^* \approx \arg\max_{\mathrm{m}_l} (\mathbb{1} - \mathrm{m}_l) \big[ \sum_{x \in \mathcal{D}_{\mathrm{r}}} \nabla_{\mathrm{m}} \ell(\mathbb{1}; \theta^*, x) \big]_l + \frac{1}{2}(\mathbb{1} - \mathrm{m}_l)^2 \big[ \widehat{\mathcal{I}}(\mathbb{1}; \theta^*, \mathcal{D}_{\mathrm{f}}) \big]_l. \tag{18}$$

where $l$ represents the layer being optimized. By leveraging the identified key modules, we can facilitate the unlearning process (e.g. GA, NPO) more effectively.

### B.2  IDENTIFY INFLUENCE-CRITICAL PARAMETERS IN MODULES

In our approach, the mask is applied to specific heads and filters, resulting in a relatively coarse granularity for unlearning. To achieve a more refined and precise method, we further examine the importance of individual parameters within these selected heads and filters. Our hypothesis is that focusing on individual parameters can help identify finer-grained regions critical for effective unlearning. To implement this, we utilize Wanda (Sun et al. (2023)) as our selection mechanism. Wanda analyzes the forget dataset, which serves as input for the selection process, and assigns importance scores to each neuron, with higher scores indicating greater relevance to the unlearning task. Based on these scores, we apply a

Figure 5: Overall results of unlearning performance are presented using MAPE-SO on BERT-base, both with and without the neuron selection mechanism. For clarity, MAPE-SO denotes MAPE-SO applied to modules at 90% sparsity, while MAPE-SO(90%) indicates MAPE-SO applied to both modules and parameters, each with 90% sparsity.

| Datasets | Method | Efficacy | | Fidelity | |
|---|---|---|---|---|---|
| | | Unlearn Scores ↓ | MIA ↓ | Remain Scores ↑ | Test Acc. ↑ |
| MNLI | MAPE-SO | 85.94% | 0.7969 | **94.15%** | **84.62%** |
| | MAPE-SO(90%) | 85.94% | 0.7969 | 94.12% | 84.61% |
| QQP | MAPE-SO | 92.19% | 0.9062 | **98.03%** | **90.72%** |
| | SU(90%) | 92.19% | **0.8828** | 97.67% | 90.46% |
| SST-2 | MAPE-SO | 94.53% | **0.8984** | **98.93%** | 93.35% |
| | MAPE-SO(90%) | 94.53% | 0.9141 | 98.92% | 93.35% |
| SQuAD v1.1 | MAPE-SO | **85.74%** | **0.5781** | **94.25%** | **87.60%** |
| | MAPE-SO(90%) | 86.16% | 0.5859 | 93.98% | 87.19% |
| SQuAD v2.0 | MAPE-SO | 77.40% | 0.6563 | **93.90%** | 73.57% |
| | MAPE-SO(90%) | 77.40% | 0.6563 | 93.75% | **73.73%** |

90% sparsity to MAPE-SO, retaining only
the most critical parameters for unlearning.
This targeted approach aims to enhance the precision of unlearning by focusing on MAPEcific
neurons within the model. Detailed results of this mechanism are provided in Table 5.

However, our experimental results reveal that incorporating the parameter selection mechanism does
not improve unlearning performance in MAPE-SO. We hypothesize that this outcome is due to the
inherent complexity of balancing unlearning precision with model utility. While selecting individual
parameters based on their Wanda scores offers a more targeted and theoretically precise unlearning
process, this fine-grained approach may unintentionally compromise the model's overall adaptability
and robustness.

## C   Unlearning Algorithm Details

**Gradient Ascent (GA).** The approach aims to achieve unlearning by maximizing the loss on the
forget dataset, thereby causing the predictions to deviate from the original data. Specifically, given a
loss function $\ell$ and the forget dataset $\mathcal{D}_f$, the goal can be expressed as follows:

$$\mathcal{L}_{\text{GA}} = -\mathbb{E}_{(x,y)\in\mathcal{D}_f}[\ell(y|x;\theta)].$$

**Gradient Difference (GD).** Maximizing the loss on the forget dataset in GA significantly degrade
model performance. To address this, the method introduces an additional loss term on the retain
dataset to maintain model performance. Specifically, given the loss function $\ell$, forgot dataset $\mathcal{D}_f$, and
retain dataset $\mathcal{D}_r$, the objective can be defined as follows:

$$\mathcal{L}_{\text{GD}} = -\mathbb{E}_{(x,y)\in\mathcal{D}_f}[\ell(y|x;\theta)] + \mathbb{E}_{(x,y)\in\mathcal{D}_r}[\ell(y|x;\theta)].$$

**Negative Preference Optimization (NPO).** Both GD and GA methods aim to maximize the loss on
the forget dataset, which can lead to catastrophic collapse. To overcome this, the method reframes
the unlearning problem as a preference optimization problem, ensuring that the predictions of the
unlearned model deviate significantly from the original model's predictions. Specifically, given the
forgot dataset $\mathcal{D}_f$, the objective can be defined as follows:

$$\mathcal{L}_{\text{NPO}} = -\frac{2}{\beta}\mathbb{E}_{(x,y)\in\mathcal{D}_f}[\log\sigma(-\beta\log\frac{\pi_\theta(y|x)}{\pi_{\text{ref}}(y|x)})],$$

where $\sigma(t) = 1/(1+e^{-t})$ is the sigmoid function, $\beta$ is the inverse temperature, and $\pi_{\text{ref}}$ is a reference
model.

**Direct Preference Optimization (DPO).** This method reparametrizes the reward function in human
feedback reinforcement learning (RLHF) and directly learns the policy from preference data. To
achieve unlearning, we replace the original responses with 'I don't know' as the new response.
Specifically, we create a dataset $\mathcal{D}_{\text{idk}}$ that includes 'I don't know', with the objective defined as
follows:

$$\mathcal{L}_{\text{DPO}} = -\frac{1}{\beta}\mathbb{E}_{(x,y,y_{\text{idk}})\in\mathcal{D}_{\text{idk}}}[\log\sigma(\beta\log\frac{\pi_\theta(y_{\text{idk}}|x)}{\pi_{\text{ref}}(y_{\text{idk}}|x)} - \beta\log\frac{\pi_\theta(y|x)}{\pi_{\text{ref}}(y|x)})].$$

## D   Additional Experimental Details and Results

### D.1   Configurations

For the traditional tasks, we randomly select the forget and retain datasets using seeds 16 or 42. We
fine-tune the models on these datasets using the AdamW optimizer. The learning rates are chosen
from $1 \times 10^{-5}, 2 \times 10^{-5}, 3 \times 10^{-5}, 5 \times 10^{-5}$. The BERT-base model is fine-tuned for 5 epochs,
while the RoBERTa-large model is fine-tuned for 10 epochs. For the TOFU task, we directly use
the fine-tuned version of the LLama2-7b-chat model, as provided in the TOFU benchmark. In
addition, the TOFU benchmark is divided into three distinct unlearning scenarios, corresponding
to small, medium, and large batch unlearning. For the WMDP task, the goal is to mitigate harmful
knowledge in existing models. To achieve this, we use the original Zephyr-7B-beta model without
further fine-tuning. This task focuses on unlearning hazardous knowledge related to biology and
cybersecurity. All experiments are conducted on two NVIDIA RTX A6000 GPUs.

## D.2 HYPERPARAMETERS

- **Traditional Tasks.** The unlearning rates for SO are selected through a grid search within the range $[1 \times 10^{-8}, 1 \times 10^{-7}]$. For MAPE-SO with 90% sparsity, the unlearning rates are chosen via grid search in the range $[8 \times 10^{-8}, 8 \times 10^{-7}]$. All methods with fine-tuning are conducted over 3 epochs. The learning rates for these methods are grid-searched between $[1 \times 10^{-5}, 5 \times 10^{-5}]$, while the learning rates module-aware unlearning are grid-searched between $[1 \times 10^{-4}, 5 \times 10^{-4}]$.

- **TOFU Tasks.** For all experiments, the batch size is set to 16. In the unlearning experiments, 5 epochs are adopted, while 10 epochs are employed in the relearning experiments. For full unlearning, the learning rate for GA and GD is determined through grid search within the range $[2 \times 10^{-6}, 2 \times 10^{-5}]$, while for NPO and DPO, the learning rate is searched within the range $[1 \times 10^{-5}, 2 \times 10^{-4}]$. For module-aware unlearning with 90% sparsity, the learning rate for GA and GD is searched within the range $[1 \times 10^{-5}, 1 \times 10^{-4}]$, and for NPO and DPO, the learning rate is searched within the range $[5 \times 10^{-5}, 3 \times 10^{-4}]$. The learning rate during the relearning process is set to $1 \times 10^{-5}$.

- **WMDP Tasks.** For all experiments, we limit the unlearning process to 150 steps, using a batch size of 4. For GA unlearning, we use a learning rate of $1.5 \times 10^{-7}$ for full unlearning and $6 \times 10^{-7}$ for module-aware unlearning with 90% sparsity. For GD unlearning, we use a learning rate $1.5 \times 10^{-7}$ for full unlearning and $5 \times 10^{-7}$ for module-aware unlearning with 90% sparsity. For NPO unlearning, we use a learning rate of $6 \times 10^{-6}$ for full unlearning and $2 \times 10^{-5}$ for module-aware unlearning with 90% sparsity.

Table 4: Overall results of unlearning performance using different unlearning methods under two fine-tuned models on MNLI dataset.

| Model | Method | Efficacy | | Fidelity | |
|---|---|---|---|---|---|
| | | Unlearn Acc. | MIA | Retain Acc. ↑ | Test Acc. ↑ |
| BERT | RT | 85.16% | 0.7500 | 97.95% | 84.78% |
| | SA | 89.84% | 0.8437 | **92.77%** | 82.05% |
| | SO | 85.94% | 0.8047 | 94.07% | 84.60% |
| | SURE-SO | 85.94% | **0.7969** | 94.11% | 84.60% |
| | MAPE-SO | 85.94% | **0.7969** | **94.15%** | **84.62%** |
| | GA | 86.72% | 0.7500 | 93.30% | 84.61% |
| | SURE-GA | 86.72% | **0.7891** | 93.11% | 84.55% |
| | MAPE-GA | **85.94%** | 0.7969 | **93.65%** | **84.68%** |
| | GD | 87.50% | 0.8125 | 93.65% | 84.52% |
| | SURE-GD | 87.50% | **0.7969** | 93.76% | 84.63% |
| | MAPE-GD | 87.50% | **0.7969** | **94.00%** | **84.65%** |
| | NPO | 86.72% | 0.8047 | **94.16%** | 84.47% |
| | SURE-NPO | 87.50% | **0.7969** | 93.67% | 84.56% |
| | MAPE-NPO | **85.94%** | **0.7969** | 93.91% | **84.71%** |
| RoBERTa | RT | 90.26% | 0.8125 | 98.79% | 90.02% |
| | SA | 92.97% | 0.8906 | 96.86% | 87.08% |
| | SO | 92.97% | **0.8906** | 94.32% | 88.99% |
| | SURE-SO | **92.19%** | 0.9141 | 95.69% | 89.45% |
| | MAPE-SO | **92.19%** | **0.8906** | **95.75%** | **89.52%** |
| | GA | 92.19% | 0.8672 | 95.95% | 89.55% |
| | SURE-GA | 92.19% | **0.8359** | 95.64% | 89.52% |
| | MAPE-GA | 92.19% | **0.8359** | **96.19%** | **89.58%** |
| | GD | 93.75% | 0.8906 | 95.98% | 89.38% |
| | SURE-GD | **92.97%** | **0.8828** | 96.09% | 89.34% |
| | MAPE-GD | **92.97%** | 0.8906 | **96.27%** | **89.57%** |
| | NPO | 91.41% | 0.8906 | 96.09% | 89.84% |
| | SURE-NPO | 91.41% | 0.8828 | 95.87% | 89.45% |
| | MAPE-NPO | 91.41% | **0.8750** | **96.23%** | **89.96%** |

Table 5: Overall results of unlearning performance using different unlearning methods under two fine-tuned models on QQP dataset.

| Model | Method | Efficacy | | Fidelity | |
|---|---|---|---|---|---|
| | | Unlearn Acc. | MIA | Retain Acc. ↑ | Test Acc. ↑ |
| BERT | RT | 92.97% | 0.8750 | 98.48% | 91.38% |
| | SA | 92.19% | 0.8906 | 92.52% | 88.52% |
| | SO | 92.97% | 0.9063 | 97.69% | 90.65% |
| | SURE-SO | **92.19%** | 0.9063 | 97.98% | 84.60% |
| | MAPE-SO | **92.19%** | **0.8906** | **98.03%** | **90.72%** |
| | GA | 93.75% | 0.9141 | **98.55%** | **91.14%** |
| | SURE-GA | 93.75% | **0.8984** | 96.84% | 89.83% |
| | MAPE-GA | 93.75% | **0.8984** | 97.17% | 90.15% |
| | GD | 94.53% | 0.9141 | **98.60%** | **91.15%** |
| | SURE-GD | 94.53% | 0.9063 | 98.54% | 91.11% |
| | MAPE-GD | 94.53% | **0.8984** | 98.54% | 91.12% |
| | NPO | 92.97% | 0.8906 | **98.55%** | 91.00% |
| | SURE-NPO | 92.97% | 0.8906 | 98.52% | 91.08% |
| | MAPE-NPO | **91.41%** | 0.8906 | 98.52% | **91.10%** |
| RoBERTa | RT | 91.41% | 0.8594 | 99.17% | 92.19% |
| | SA | 93.75% | 0.8750 | 98.69% | 91.48% |
| | SO | 92.97% | 0.8750 | **98.93%** | **91.56%** |
| | SURE-SO | 92.97% | 0.8750 | 97.91% | 91.01% |
| | MAPE-SO | 92.97% | 0.8750 | 98.86% | 91.46% |
| | GA | 92.97% | 0.9219 | 98.85% | 91.68% |
| | SURE-GA | 92.97% | 0.9219 | 98.70% | 91.47% |
| | MAPE-GA | 92.19% | 0.9219 | **98.89%** | **91.84%** |
| | GD | 92.19% | 0.9219 | 99.19% | 91.55% |
| | SURE-GD | 92.19% | 0.8906 | 98.70% | 91.38% |
| | MAPE-GD | 92.19% | **0.8828** | **99.45%** | **91.62%** |
| | NPO | 92.19% | 0.9219 | 99.14% | 91.50% |
| | SURE-NPO | 92.19% | **0.9063** | 98.87% | 91.26% |
| | MAPE-NPO | 92.19% | **0.9063** | **99.26%** | **91.63%** |

## D.3 EXAMINING UNLEARNING STRATEGY VARY SPARSITY

(Pochinkov & Schoots (2024)) argued that pruning filters are more effective than pruning heads. To further investigate this claim, we conducted a comparative analysis of three selective parameter update strategies: heads-only, filters-only, and heads&filters in Figure 6. All experiments were designed to provide comparable unlearning guarantees varying sparsity. The heads-only strategy consistently underperformed compared to the other strategies, highlighting its limited effectiveness. In contrast, the filters-only strategy not only maintained stability but also delivered consistently strong unlearning performance for SO. For fine-tuning-based unlearning methods, the heads-only strategy was found to be more effective at higher sparsity (70% to 90%), while the filters-only strategy demonstrated superior performance at lower sparsity (0% to 50%).

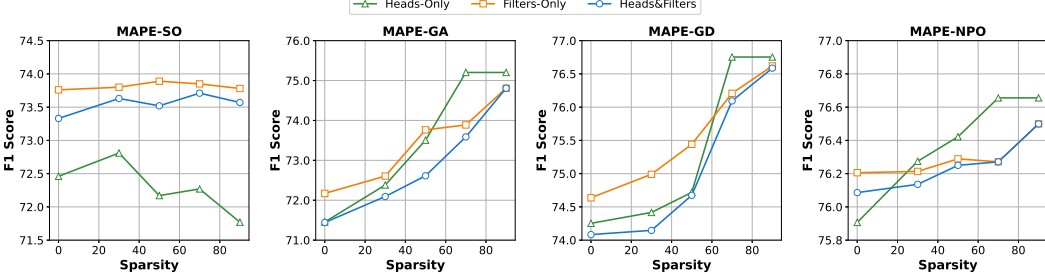

Figure 6: F1 scores for various sparsity applied to BERT-base after three kinds of unlearning strategies with different unlearning methods.

## D.4 ADDITIONAL EXPERIMENTAL RESULTS ON TRADITIONAL TASKS

We compare our approach to other unlearning methods across three classification tasks (MNLI, QQP, and SST-2) and one question-answering task (SQuAD v1.1), using two models (details provided in Table 4 to Table 7). The evaluation metrics differ by task: F1 scores are reported for the question-answering task, while accuracy is used for the classification tasks.

Table 6: Overall results of unlearning performance using different unlearning methods under two fine-tuned models on SST-2 dataset.

| Model | Method | Efficacy | | Fidelity | |
|---|---|---|---|---|---|
| | | Unlearn Acc. | MIA | Retain Acc. ↑ | Test Acc. ↑ |
| BERT | RT | 93.75% | 0.9297 | 99.06% | 93.00% |
| | SA | 95.31% | 0.9062 | 98.82% | 89.79% |
| | SO | 94.53% | 0.9141 | **98.96%** | 92.89% |
| | SURE-SO | 94.53% | 0.8984 | 98.24% | 93.12% |
| | MAPE-SO | 94.53% | 0.8984 | 98.93% | **93.35%** |
| | GA | 94.53% | 0.8125 | **96.96%** | **91.78%** |
| | SURE-GA | 94.53% | **0.7969** | 96.92% | 91.40% |
| | MAPE-GA | 94.53% | 0.8047 | 96.96% | 91.51% |
| | GD | 95.31% | 0.8359 | **97.13%** | 92.55% |
| | SURE-GD | 95.31% | **0.8672** | 96.93% | 91.97% |
| | MAPE-GD | 95.31% | **0.8672** | 97.16% | **92.66%** |
| | NPO | 93.75% | **0.8594** | **96.95%** | **91.40%** |
| | SURE-NPO | 93.75% | 0.8672 | 96.66% | 90.60% |
| | MAPE-NPO | 93.75% | 0.8672 | 96.75% | 90.94% |
| RoBERTa | RT | 94.53% | 0.9063 | 99.64% | 96.10% |
| | SA | 94.53% | 0.9219 | 98.84% | 95.07% |
| | SO | 94.53% | 0.9297 | 99.14% | 94.15% |
| | SURE-SO | 94.53% | 0.9375 | 98.94% | 94.20% |
| | MAPE-SO | 94.53% | 0.8984 | **99.45%** | **94.55%** |
| | GA | 93.75% | 0.9375 | 98.69% | 95.07% |
| | SURE-GA | 93.75% | 0.9219 | 98.92% | 95.30% |
| | MAPE-GA | 93.75% | 0.9219 | **98.96%** | **95.36%** |
| | GD | 93.75% | 0.9297 | 98.95% | 95.41% |
| | SURE-GD | 93.75% | **0.8516** | 98.87% | 95.26% |
| | MAPE-GD | 93.75% | **0.8516** | **99.00%** | **95.47%** |
| | NPO | 93.75% | 0.8672 | **99.09%** | 95.20% |
| | SURE-NPO | 93.75% | **0.8516** | 98.67% | 95.06% |
| | MAPE-NPO | 93.75% | **0.8516** | **99.09%** | **95.27%** |

Table 7: Overall results of unlearning performance using different unlearning methods under two fine-tuned models on SQuAD v1.1 dataset.

| Model | Method | Efficacy | | Fidelity | |
|---|---|---|---|---|---|
| | | Unlearn F1. | MIA | Retain F1. ↑ | Test F1. ↑ |
| BERT | RT | 87.62 | 0.5938 | 95.23 | 88.18 |
| | SA | 89.75 | 0.7031 | 91.94 | 86.85 |
| | SO | 86.26 | **0.5625** | **94.33** | **87.74** |
| | SURE-SO | 86.52 | 0.6016 | 94.08 | 87.31 |
| | MAPE-SO | 86.74 | 0.5781 | 94.25 | 87.60 |
| | GA | 88.35 | 0.6953 | 95.09 | 88.08 |
| | SURE-GA | 88.26 | **0.6875** | **95.10** | 87.95 |
| | MAPE-GA | 87.83 | **0.6875** | **95.10** | 88.09 |
| | GD | 89.17 | 0.6797 | 95.10 | **88.18** |
| | SURE-GD | 89.04 | 0.6797 | 95.11 | 88.07 |
| | MAPE-GD | 88.65 | 0.6797 | **95.14** | **88.18** |
| | NPO | 88.78 | 0.6953 | 95.06 | 88.17 |
| | SURE-NPO | 88.26 | 0.6797 | 95.03 | 88.05 |
| | MAPE-NPO | 87.87 | **0.6719** | **95.10** | 88.20 |
| RoBERTa | RT | 90.41 | 0.6484 | 97.92 | 92.50 |
| | SA | 91.05 | 0.6875 | 95.16 | 89.36 |
| | SO | 90.71 | **0.5000** | 94.93 | 90.95 |
| | SURE-SO | 90.44 | 0.5859 | 95.14 | 91.03 |
| | MAPE-SO | 90.81 | 0.5234 | **95.51** | **91.06** |
| | GA | 91.46 | 0.6953 | 96.16 | 91.01 |
| | SURE-GA | 92.32 | 0.6953 | 96.00 | 90.78 |
| | MAPE-GA | **90.94** | 0.6953 | **96.20** | **91.01** |
| | GD | 91.47 | 0.6953 | 96.58 | 91.22 |
| | SURE-GD | 91.76 | 0.6953 | 95.95 | 90.80 |
| | MAPE-GD | **91.35** | 0.6719 | **96.69** | **91.43** |
| | NPO | 91.62 | 0.6797 | 95.98 | 90.64 |
| | SURE-NPO | 91.29 | 0.6797 | 95.94 | 89.90 |
| | MAPE-NP0 | **91.12** | **0.6641** | **96.17** | 90.82 |

## D.5 ADDITIONAL EXPERIMENTAL RESULTS ON TOFU TASK

Tables 8 and 9 present additional results for the Forget01 and Forget10 unlearning scenarios in the TOFU task, respectively. We observe that our method performs less effectively when the number of forget samples is small, under-performing compared to full-parameter updates. However, as the number of forget samples increases, our method outperforms others, achieving an optimal balance between forget quality and model utility.

Table 8: Performance overview of various unlearning methods on Forget01 unlearning settings.

| Method | Efficacy | | | | Fidelity | | | | | | | | | MU ↑ |
|---|---|---|---|---|---|---|---|---|---|---|---|---|---|---|
| | Forget Set | | | FQ ↑ | Real Authors | | | World Facts | | | Retain Set | | | |
| | Rouge | Prob. | TR | | Rouge ↑ | Prob. ↑ | TR ↑ | Rouge ↑ | Prob. ↑ | TR ↑ | Rouge ↑ | Prob. ↑ | TR ↑ | |
| RT | 0.39 | 0.18 | 0.69 | 1.0 | 0.93 | 0.45 | 0.58 | 0.88 | 0.41 | 0.54 | 0.99 | 0.99 | 0.47 | 0.62 |
| SA | 0.95 | 0.99 | 0.53 | 1.88e-4 | 0.93 | 0.45 | 0.58 | 0.87 | 0.42 | 0.56 | 0.98 | 0.99 | 0.48 | 0.62 |
| GA | 0.49 | 0.23 | 0.54 | 1.27e-3 | 0.92 | 0.42 | 0.55 | **0.89** | 0.41 | 0.54 | 0.92 | 0.95 | 0.49 | 0.60 |
| SURE-GA | 0.48 | **0.17** | 0.55 | **3.02e-3** | 0.93 | 0.42 | 0.55 | 0.87 | 0.41 | 0.54 | 0.92 | 0.88 | 0.49 | 0.30 |
| MAPE-GA | **0.46** | 0.22 | **0.56** | **3.02e-3** | **0.94** | **0.43** | **0.56** | 0.86 | **0.42** | **0.55** | **0.94** | **0.97** | 0.49 | **0.61** |
| GD | 0.48 | 0.30 | 0.53 | 1.27e-3 | 0.92 | 0.42 | 0.55 | 0.87 | 0.40 | 0.54 | 0.92 | 0.97 | 0.49 | 0.60 |
| SURE-GD | 0.48 | **0.24** | 0.54 | 1.27e-3 | 0.92 | 0.42 | 0.55 | **0.88** | **0.41** | 0.54 | 0.91 | 0.94 | 0.49 | 0.60 |
| MAPE-GD | **0.47** | 0.28 | **0.55** | **3.02e-3** | 0.92 | **0.43** | **0.56** | 0.86 | **0.41** | **0.55** | 0.92 | **0.98** | 0.49 | **0.61** |
| DPO | 0.47 | **0.85** | **0.60** | **6.76e-3** | 0.94 | **0.49** | 0.63 | 0.87 | **0.46** | 0.57 | **0.89** | 0.96 | 0.46 | 0.63 |
| SURE-DPO | 0.48 | 0.87 | 0.58 | 3.02e-3 | 0.94 | 0.48 | 0.62 | 0.87 | 0.45 | 0.57 | 0.88 | 0.96 | 0.46 | 0.63 |
| MAPE-DPO | **0.44** | **0.85** | 0.59 | 3.02e-3 | 0.94 | 0.48 | **0.63** | 0.87 | 0.45 | 0.57 | 0.88 | 0.96 | **0.47** | 0.63 |
| NPO | 0.45 | **0.14** | 0.59 | 6.76e-3 | **0.94** | 0.41 | 0.54 | 0.87 | 0.40 | 0.53 | 0.90 | 0.85 | 0.49 | 0.59 |
| SURE-NPO | 0.45 | 0.07 | **0.64** | **0.27** | 0.93 | 0.41 | 0.53 | 0.87 | 0.40 | 0.52 | 0.89 | 0.85 | 0.49 | 0.59 |
| MAPE-NPO | **0.44** | 0.07 | **0.64** | **0.27** | 0.93 | **0.42** | **0.55** | **0.89** | 0.41 | **0.54** | 0.91 | 0.85 | 0.49 | **0.60** |

Table 9: Performance overview of various unlearning methods on Forget10 unlearning settings.

| Method | Efficacy | | | | Fidelity | | | | | | | | | MU ↑ |
|---|---|---|---|---|---|---|---|---|---|---|---|---|---|---|
| | Forget Set | | | FQ ↑ | Real Authors | | | World Facts | | | Retain Set | | | |
| | Rouge | Prob. | TR | | Rouge ↑ | Prob. ↑ | TR ↑ | Rouge ↑ | Prob. ↑ | TR ↑ | Rouge ↑ | Prob. ↑ | TR ↑ | |
| RT | 0.41 | 0.15 | 0.67 | 1.0 | 0.93 | 0.43 | 0.57 | 0.90 | 0.41 | 0.54 | 0.98 | 0.99 | 0.47 | 0.61 |
| SA | 0.98 | 0.99 | 0.50 | 1.69e-15 | 0.92 | 0.44 | 0.58 | 0.86 | 0.41 | 0.55 | 0.98 | 0.99 | 0.49 | 0.62 |
| GA | 0.42 | 0.04 | **0.54** | **7.28e-9** | 0.87 | 0.37 | 0.51 | **0.87** | 0.37 | 0.51 | 0.44 | 0.09 | 0.46 | 0.33 |
| SURE-GA | 0.42 | 0.03 | 0.51 | 9.25e-11 | 0.82 | 0.38 | **0.53** | 0.86 | 0.39 | **0.54** | 0.45 | 0.07 | 0.46 | 0.30 |
| MAPE-GA | 0.42 | **0.20** | 0.53 | 8.78e-12 | **0.89** | 0.39 | 0.53 | 0.84 | **0.40** | 0.53 | **0.48** | **0.46** | 0.46 | **0.51** |
| GD | **0.41** | **0.15** | 0.49 | 5.56e-14 | 0.83 | 0.40 | 0.56 | **0.87** | 0.38 | 0.52 | 0.49 | 0.55 | 0.48 | 0.52 |
| SURE-GD | **0.41** | 0.20 | 0.49 | **3.92e-13** | 0.85 | 0.41 | 0.55 | 0.86 | 0.38 | 0.51 | **0.55** | **0.67** | **0.50** | **0.54** |
| MAPE-GD | 0.40 | 0.17 | **0.50** | **3.92e-13** | 0.86 | **0.42** | **0.57** | 0.86 | **0.39** | **0.52** | 0.51 | 0.59 | 0.50 | 0.54 |
| DPO | 0.25 | 0.26 | 0.68 | **0.02** | 0.58 | 0.43 | 0.55 | 0.79 | **0.43** | 0.55 | 0.45 | 0.62 | 0.42 | 0.51 |
| SURE-DPO | 0.27 | **0.19** | 0.64 | 9.06e-4 | 0.69 | **0.46** | **0.59** | 0.75 | 0.42 | 0.54 | **0.57** | **0.79** | 0.45 | **0.56** |
| MAPE-DPO | **0.28** | 0.20 | 0.64 | 3.11e-3 | **0.81** | 0.39 | 0.49 | **0.85** | 0.41 | 0.52 | 0.55 | 0.77 | **0.45** | 0.54 |
| NPO | **0.27** | **0.11** | 0.72 | 3.36e-2 | 0.72 | **0.46** | **0.62** | **0.86** | 0.45 | **0.59** | 0.35 | 0.29 | 0.36 | 0.47 |
| SURE-NPO | 0.25 | 0.07 | **0.68** | 0.45 | **0.73** | **0.46** | 0.60 | **0.86** | 0.45 | 0.57 | 0.44 | 0.60 | 0.43 | **0.54** |
| MAPE-NPO | 0.25 | 0.06 | **0.66** | **0.90** | 0.69 | 0.45 | 0.57 | 0.84 | 0.45 | **0.59** | 0.45 | 0.61 | 0.44 | **0.54** |

## D.6 ADDITIONAL EXPERIMENTAL RESULTS ON HAZARDOUS KNOWLEDGE REMOVAL TASK

Figure 7: Performance comparison of unlearning methods on WMDP under Zephyr-7B-beta.

| Method | Efficacy | | | Fidelity |
|---|---|---|---|---|
| | AccBio. ↓ | AccCyber. ↓ | Avg. ↓ | MMLU ↑ |
| Original | 0.6465 | 0.4449 | 0.5457 | 0.5845 |
| GA | 0.2679 | 0.3301 | 0.2990 | 0.4083 |
| SURE-GA | **0.2569** | 0.3296 | **0.2933** | 0.4030 |
| MAPE-GA | 0.2726 | **0.3191** | 0.2959 | **0.4145** |
| GD | 0.3370 | 0.3709 | 0.3540 | 0.4529 |
| SURE-GD | 0.3346 | 0.3749 | 0.3548 | **0.4603** |
| MAPE-GD | **0.3236** | **0.3629** | **0.3433** | 0.4557 |
| NPO | 0.4540 | 0.4051 | 0.4396 | 0.4903 |
| SURE-NPO | 0.4588 | 0.3905 | 0.4247 | 0.4900 |
| MAPE-NPO | **0.4454** | **0.3900** | **0.4177** | **0.4930** |