# OpenReview forum: "Module-Aware Parameter-Efficient Machine Unlearning on Transformers"
_ICLR.cc/2026/Conference — ICLR 2026 Conference Withdrawn Submission_

### Official Review · Reviewer_2FGw · 2025-10-30

**Soundness:** 3
**Presentation:** 3
**Contribution:** 3
**Rating:** 6
**Confidence:** 3

**Summary:**

The paper proposes MAPE-Unlearn, a module-aware parameter-efficient unlearning method for Transformer-based models. The key idea is to learn masks to pinpoint influence-critical parameters in the heads and filters of Transformers. These masks are derived from an objective that combines gradients on the forget set and Fisher/Hessian on the retain set. These learned masks are then integrated into both second-order updates and fine-tuning-based unlearning. Experiments include traditional classification tasks and QA tasks under GLUE, SQuAD; unlearning tasks under TOFU and WMDP.

**Strengths:**

1. The paper addresses an important problem: unlearning in large Transformer models. The idea of module-level selection is well-motivated by the Transformer structure.
2. The paper writing is clear.
3. The empirical evaluation is comprehensive. Extensive experiments on diverse tasks using different models demonstrate that the proposed method achieves strong forgetting while preserving utility.

**Weaknesses:**

1.    The approach assumes continued access to both the forget set and the retain set in order to compute gradients and Fisher/Hessian. This is a strong assumption. Large LLMs often cannot reconstruct a clean retain set. The paper does not clarify how MAPE-Unlearn applies when forget data $D_f$ or retain data $D_r$ is unavailable.
2.    Although the method is called “parameter-efficient,” the mask computation itself is still expensive. This cost must be paid for each unlearning request. The paper does not quantify compute/memory overhead, nor does it provide guarantees on when the greedy, block-diagonal Fisher–based approximation will select the right modules. As a result, the scalability to very large models and long-term maintenance remain questionable.
3.    The method’s reliability at the extremes is unclear.  If the forget set is extremely small (e.g., one data point), the gradient may be too small, so the model may fail to remove that data. If the forget set is extremely large (e.g., 30% of the data set), the “influence-critical” region may no longer be sparse. The paper does not evaluate these regimes, so it is unclear how robust MAPE-Unlearn is.
4.    The paper lacks a theoretical guarantee that the learned masks actually remove the influence of the forget data.

**Questions:**

Please see the Weaknesses section for all questions and clarification requests.

---

### Official Review · Reviewer_BH1H · 2025-11-01

**Soundness:** 2
**Presentation:** 1
**Contribution:** 2
**Rating:** 2
**Confidence:** 4

**Summary:**

This paper introduces MAPE-Unlearn, a module-aware parameter-efficient unlearning method for Transformers. Unlike existing approaches that identify critical parameters at fine-grained levels, MAPE-Unlearn employs learnable binary masks at the module level (attention heads and feed-forward filters) to pinpoint influence-critical parameters more effectively. The mask optimization leverages Fisher Information Matrix and greedy search with warm start. MAPE-Unlearn integrates seamlessly with various unlearning methods (e.g., SO, GA, NPO). Experiments across traditional NLP tasks, TOFU benchmark, and hazardous knowledge removal demonstrate its effectiveness at 90% sparsity, achieving superior balance between unlearning efficacy and model fidelity. The method also shows enhanced robustness in successive unlearning scenarios and against relearning attacks.

**Strengths:**

1.	The method uniquely operates at the module level (heads and filters) rather than individual parameters, better capturing Transformer architecture patterns.
2.	Comprehensive experiments across diverse tasks and models demonstrate superior effectiveness at 90% sparsity while maintaining model fidelity.

**Weaknesses:**

1.	While the method is empirically strong, the paper provides limited theoretical justification for why module-level unlearning should be fundamentally more effective than parameter-level approaches for Transformers. The connection between module structure and unlearning efficacy remains somewhat heuristic.
2.	The paper lacks thorough ablation studies on mask initialization strategies and the warm-start greedy algorithm. It remains unclear how sensitive the results are to these design choices, or if simpler approaches could achieve comparable performance.
3.	Experiments are confined to encoder-style (BERT, RoBERTa) and some decoder-style LLMs (LLaMA, Zephyr), but don't include encoder-decoder models (like T5) or vision transformers, leaving generalizability across full Transformer architectures unverified.

**Questions:**

1.	The paper claims module-aware masks provide "tighter bounded approximation error" for second-order unlearning. Is this bound derived theoretically or observed empirically? If theoretical, please state the bound; if empirical, how was the approximation error directly measured?
2.	The core premise is that module-level interactions are crucial for unlearning in Transformers. Beyond the improved results, what direct evidence or analysis (e.g., attention pattern visualization, feature space analysis) confirms that the identified masks indeed correspond to functionally coherent modules responsible for the "forgotten" knowledge?
3.	The diagonal Fisher approximation is central to your efficient optimization. However, in highly correlated modules like attention heads, do significant off-diagonal elements exist whose omission limits the theoretically achievable performance? Did you experiment with block-diagonal or low-rank approximations?
4.	The 90% sparsity ratio is consistently presented as optimal. Was this ratio held fixed across all models (from BERT-base to LLaMA2-7B)? Given the vast differences in model architecture and scale, shouldn't the optimal sparsity be model-dependent? How was the potential for overfitting to this specific value addressed?

---

### Official Review · Reviewer_RsXX · 2025-11-01

**Soundness:** 2
**Presentation:** 2
**Contribution:** 2
**Rating:** 4
**Confidence:** 4

**Summary:**

This paper introduces MAPE-Unlearn, a module-aware parameter-efficient unlearning framework for Transformer models. Unlike previous module-oblivious approaches that fail to identify influence-critical components, MAPE-Unlearn targets functional modules—attention heads and feed-forward filters—via learnable masks optimized through a warm-start greedy search. This module-level masking strategy can be integrated with existing unlearning paradigms such as second-order and gradient-ascent unlearning. Experiments across BERT, RoBERTa, Llama2, and Zephyr show that MAPE-Unlearn achieves a better balance between unlearning effectiveness and model fidelity, maintaining robustness even under high sparsity, successive unlearning, and relearning attacks.

**Strengths:**

1. The paper’s key strength lies in introducing a module-aware unlearning paradigm that aligns with the inherently modular Transformer architecture. By targeting attention heads and feed-forward filters, it effectively addresses the limitations of prior parameter-efficient methods (e.g., SA, SURE) that overlook modularity or focus on fine-grained pruning.
2. The method is theoretically grounded rather than heuristic. The learnable masks are derived from unlearning objectives (MLR, MLF) that integrate “forget gradients” with the “retain FIM,” and the diagonal FIM approximation provides a good balance between accuracy and efficiency for large-scale models.
3. The experiments are extensive and convincing, covering diverse tasks (GLUE, SQuAD, TOFU, WMDP), models (BERT, Llama2, Zephyr), and unlearning algorithms (GA, GD, NPO, SO), demonstrating strong generalization and applicability.

**Weaknesses:**

1. As noted in Appendix D.5, the method underperforms full-parameter updates when the number of forget samples is small (e.g., TOFU Forget01). Since many real-world unlearning requests involve limited data, this limitation is particularly noteworthy.
2. The authors fix the sparsity level at 90 percent for most experiments but do not analyze how varying sparsity (e.g., 50%, 70%, 95%) affects performance across tasks, leaving scalability under different sparsity settings unexplored.

**Questions:**

The rationale for selecting attention heads and feed-forward filters as target modules requires clearer justification. The paper should explain why these components are prioritized over others such as LayerNorm, residual connections, or positional encodings, and whether ablation studies were conducted to compare their unlearning efficacy. It also remains unclear if heads and filters consistently serve as the most influence-critical modules across diverse tasks like QA and sentiment analysis, or if optimal module types vary by task. Clarifying this would determine whether MAPE-Unlearn’s module-aware design is task-agnostic or task-specific.

Additionally, the method’s adaptability to small-scale forget datasets, such as TOFU Forget01, needs further discussion. Since performance drops when the forget set is small, it would be helpful to know whether adaptive strategies—such as increasing the weight of the forget gradient term or lowering sparsity to allow broader parameter updates—have been explored. Evaluating MAPE-Unlearn under extremely limited data (<0.5% of training samples) against parameter-efficient baselines like SA or SURE would further demonstrate its robustness in realistic low-data unlearning scenarios.

---

### Official Review · Reviewer_Wxqn · 2025-11-03

**Soundness:** 2
**Presentation:** 2
**Contribution:** 2
**Rating:** 2
**Confidence:** 4

**Summary:**

This paper proposes MAPE-Unlearn, a module-aware and parameter-efficient framework for machine unlearning in Transformer models. Rather than updating all parameters or pruning individual weights, it selectively adjusts only a small set of influential modules using binary masks. These masks are optimized via a second-order Taylor approximation with a diagonal Fisher Information Matrix, allowing accurate influence estimation while keeping computation lightweight. The framework is compatible with common unlearning objectives such as SO, GA, GD, and NPO, producing variants like MAPE-SO and MAPE-GA. Overall, the method demonstrates that sparse, module-level updates enable efficient, scalable, and reliable unlearning in large language models.

**Strengths:**

1.	The paper introduces a novel module-aware masking strategy that updates only influential attention heads and feed-forward filters. This design reduces computational cost while preserving unlearning effectiveness.
2.	Across GLUE, SQuAD, TOFU, and hazardous knowledge tasks, MAPE-Unlearn consistently outperforms baselines by achieving effective forgetting with minimal degradation in model fidelity.
3.	The method shows clear advantages under successive unlearning and relearning attacks—two realistic and under-explored scenarios—demonstrating better stability, lower error accumulation, and higher resistance to knowledge recovery.

**Weaknesses:**

1.	The discussion of PEFT-related unlearning methods is not up-to-date and omits several recent works such as [1-3]. This makes the positioning of the paper less convincing and weakens its contribution relative to current literature.
2.	The paper does not clearly articulate why unlearning in Transformers poses unique challenges for PEFT-based methods. It also fails to explicitly map these challenges to its proposed module-aware solution, making the motivation appear incomplete and insufficiently grounded.
3.	Although extensive experiments are conducted, the reliance on influence-function-based Taylor expansion for module selection feels unnecessary, given recent unlearning methods [1] that efficiently update adapter parameters using Hessian–vector product approximations without sacrificing utility. Moreover, the paper inaccurately implies direct inverse-Hessian computation, rather than recognizing existing efficient approximations.
4.	The paper does not provide empirical results on the time/latency of parameter (module) selection and update processes. It also lacks visualization or reporting of the selected module groups, making it difficult to verify whether the chosen modules are meaningful or consistent across runs.
[1] Ding, C., Wu, J., Yuan, Y., Lu, J., Zhang, K., Su, A., ... & He, X. (2024). Unified parameter-efficient unlearning for llms. arXiv preprint arXiv:2412.00383.
[2] Huo, J., Yan, Y., Zheng, X., Lyu, Y., Zou, X., Wei, Z., & Hu, X. (2025). Mmunlearner: Reformulating multimodal machine unlearning in the era of multimodal large language models. arXiv preprint arXiv:2502.11051.
[3] Liu, Z., Dou, G., Yuan, X., Zhang, C., Tan, Z., & Jiang, M. (2025). Modality-aware neuron pruning for unlearning in multimodal large language models. arXiv preprint arXiv:2502.15910.

**Questions:**

What is the error analysis and theoretical justification for using the diagonal Fisher Information Matrix as an approximation? Under what conditions does this approximation fail or become invalid?

---

### Note · Authors · 2025-11-28

I have read and agree with the venue's withdrawal policy on behalf of myself and my co-authors.